# The Usefulness of Tissue Calprotectin in Pediatric Crohn's Disease—A Pilot Study

Edyta Szymanska [1,*], Sylwia Szymanska [2], Agnieszka Karkucińska-Więckowska [2], Aldona Wierzbicka [3], Jaroslaw Kierkus [1] and Maciej Dadalski [1]

1   Department of Gastroenterology, Hepatology, Feeding Disorders and Pediatrics, Children's Memorial Health Institute, 04-736 Warsaw, Poland
2   Department of Patomorphology, Children's Memorial Health Institute, 04-736 Warsaw, Poland
3   Department of Biochemistry and Experimental Medicine, Children's Memorial Health Institute, 04-736 Warsaw, Poland
*   Correspondence: edyta.szymanska@onet.com.pl

**Abstract: Background:** Fecal calprotectin (FCP) is a highly sensitive biomarker of intestinal inflammation widely used in diagnostics and monitoring of inflammatory bowel disease (IBD). Immunohistochemical assessment of calprotectin in the bowel mucosa is not a diagnostic standard. Therefore, the aim of this study was to evaluate tissue calprotectin (TCP) as a potential marker providing added insight for pediatric patients with Crohn's disease (CD). **Methods**: Fecal and tissue calprotectin were measured in children with CD. The values were correlated with disease activity and histopathological changes of the patients' endoscopic biopsies. Disease activity was assessed using the Pediatric Crohn's Disease Activity Index (PCDAI); fecal calprotectin (FCP) was measured with the ELISA test. Immunohistochemical (IHC) staining for calprotectin antigen was performed on the biopsy samples from six bowel segments, and the number of TCP cells was counted per high power field (HPF). Non-parametric statistical tests were used for data analysis. **Results**: Fifty-seven children with CD with a median age of 10.5 (1–17) years (yrs) were examined for fecal and tissue calprotectin. The patients' median PCDAI score was 10 (0–63.5), while median FCP was 535 (30–600) μg/g. We observed a correlation between disease activity (PCDAI) and FCP, TCP in inflammatory lesions and in crypts. There was no association either between FCP and TCP or between TCP in epithelium and PCDAI. **Conclusion**: It seems that IHC detection of calprotectin in bowel mucosa to assess disease behavior may be useful. FCP is a gold-standard biomarker in the diagnosis, monitoring and prognosis of IBD, and its levels correlated well with clinical activity in our study group.

**Keywords:** fecal calprotectin; tissue calprotectin; biomarkers; inflammatory bowel disease

## 1. Introduction

Crohn's disease (CD) is a chronic gastro-intestinal condition with impaired immunological background [1]. Diagnosis of CD is based on clinical symptoms, laboratory parameters and histopathological examination of the endoscopic intestinal biopsies [2,3]. However, for a long time now, non-invasive biomarkers of intestinal inflammation have been widely used in clinical practice. They allow discrimination between functional and organic bowel disorders and are used not only in diagnosis but also in disease monitoring [4].

So far, fecal calprotectin (FCP) is the most popular and common biomarker in IBD and the only one officially approved by the European Crohn's and Colitis Organization (ECCO) [2,3].

Calprotectin is an acute phase reactant protein, which belongs to the S100 leukocyte protein family and is found in the granulocytes, neutrophil cytosol, in monocytes and activated macrophages [5]. When released into the extracellular space, it induces migration of neutrophils to the inflammatory lesion, stimulates their phagocytic activity and induces apoptosis, both in normal and cancerous cells [6,7]. Since intestinal inflammation leads to a

significant increase in calprotectin in the stool, it has been used as a marker of inflammatory processes in the gastro-intestinal tract, such as IBD [8,9].

We need to remember, however, that histopathological features play a crucial role in predicting disease behavior. Active inflammation and myenteric plexitis were found to be the potential predictors of the disease course, while chronic inflammation and granulomas in the resection margin did not show such correlation [10–12]. It has been reported that S100-positive enteric glial cells were associated with both clinical and endoscopic recurrence [13].

The immunohistochemical (IHC) detection of calprotectin is not a standard method either for monitoring or predicting the disease course or its prognosis. There are also limited data on the usefulness of tissue calprotectin (TCP) in IBD. Moreover, the available results from the pilot studies are not consistent. On the one hand, higher levels of TCP have been observed in IBD than in control groups, while its lower levels have been detected in patients with ulcerative colitis (UC) in remission [14–16]. On the other hand, there are studies demonstrating no correlation between TCP and IBD course [17,18].

The aim of this study was to evaluate TCP as a biomarker providing added insight for pediatric patients with CD.

## 2. Results

### 2.1. General Informatins

The study included 57 pediatric patients with CD with a median age of 10.5 (1–17) years (yrs). The patients' median PCDAI score was 10 (0–63.5), which indicates a mild–moderate course of the disease. The median FCP level was 535 (30–600) μg/g. Table 1 presents the patient characteristics.

**Table 1.** Characteristics of the study group, including clinical, laboratory, endoscopic and histopathological data.

| Factor/Parameter | Total Cohort (N = 57) |
|---|---|
| Boys | 32 (56%) |
| Age yrs (median) | 10.5 (1–17) |
| FCP (μg/g)/median | 535 (30–600) μg/g |
| PCDAI | 10 (0–63.5) |
| SES-CD median | 12 (0–32) |
| CRP (mg/g) N < 0.5 | 0.3 (0.02–7.3) |
| Hemoglobin (g/dL) median | 12.1 (9.8–17.2) |
| BMI median | 16.6 (9.4–22.6) |
| Medication | |
| - 5-ASA | - 27 (47.4%) |
| - AZA/MTX | - 45 (79%) |
| - GKS | - 15 (26.3%) |
| - IFX | - 50 (87.7) |
| **TCP (0–3)/mean** | |
| - infiltration | 0.6 |
| - epithelium | 0.19 |
| - crypts | 0.28 |

FCP—fecal calprotectin; TCP—tissue calprotectin; PCDAI—Pediatric Crohn's Disease Activity Index; SES-CD—simple endoscopic scale for CD; Yrs—years; CRP—C-reactive protein; BMI—body mass index; 5-ASA—5-aminosalicylate; GKS—glycorticosteroids.

### 2.2. Histopathological Features

Three patterns of calprotectin staining were observed (picture A and B): in inflammatory infiltrates, within the surface epithelium and crypt epithelium. The most common

pattern was the one in inflammatory infiltrates (observed in 26 patients). However, it was mostly faint and focal. Staining was strong in only one patient. Surface pattern was observed in 7 patients, while positivity in crypt epithelium was diagnosed in 15 patients, respectively.

The correlation between FCP and clinical disease activity (PCDAI) was reported (r-Pearson: 0.31, r-spearman–0.186, p-0.16); the higher the PCDAI score (more active disease), the higher the FCP level (Figure 1). We also observed the correlation between PCDAI and TCP in inflammatory infiltration (p-0.003) and in crypts (0.025) (Figures 2–4, respectively). No association was observed either between FCP and TCP (r-Pearson: 0.139) or between FCP and TCP in epithelium (r-Pearson: 0.053, p-0.34). In some cases, even the opposite relationship was observed between PCDAI and TCP in epithelium—the lower the PCDAI (milder disease), the higher the TCP in epithelium.

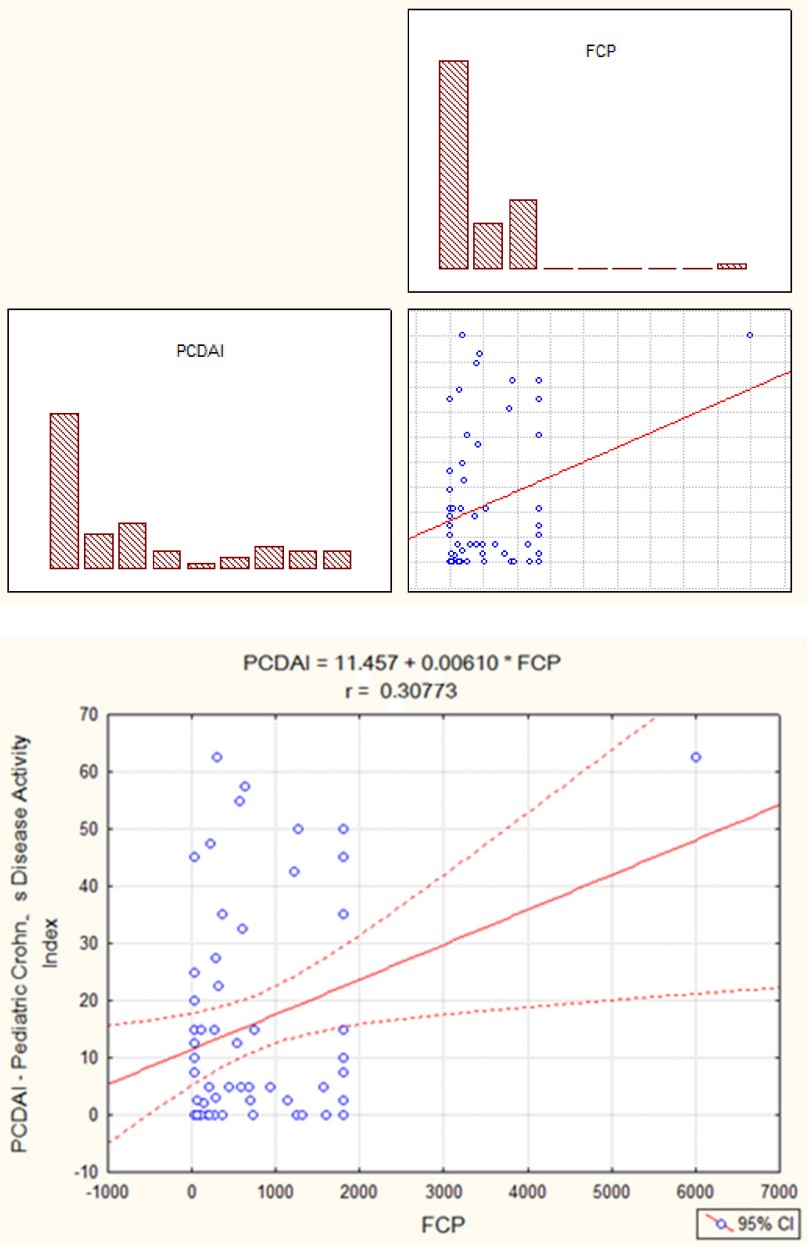

**Figure 1.** The level of FCP and clinical disease activity (PCDAI) of 57 patients included in the study; the higher the PCDAI score (more active disease), the higher the FCP level (r-Pearson: 0.31, r-spearman–0.186, p-0.16).

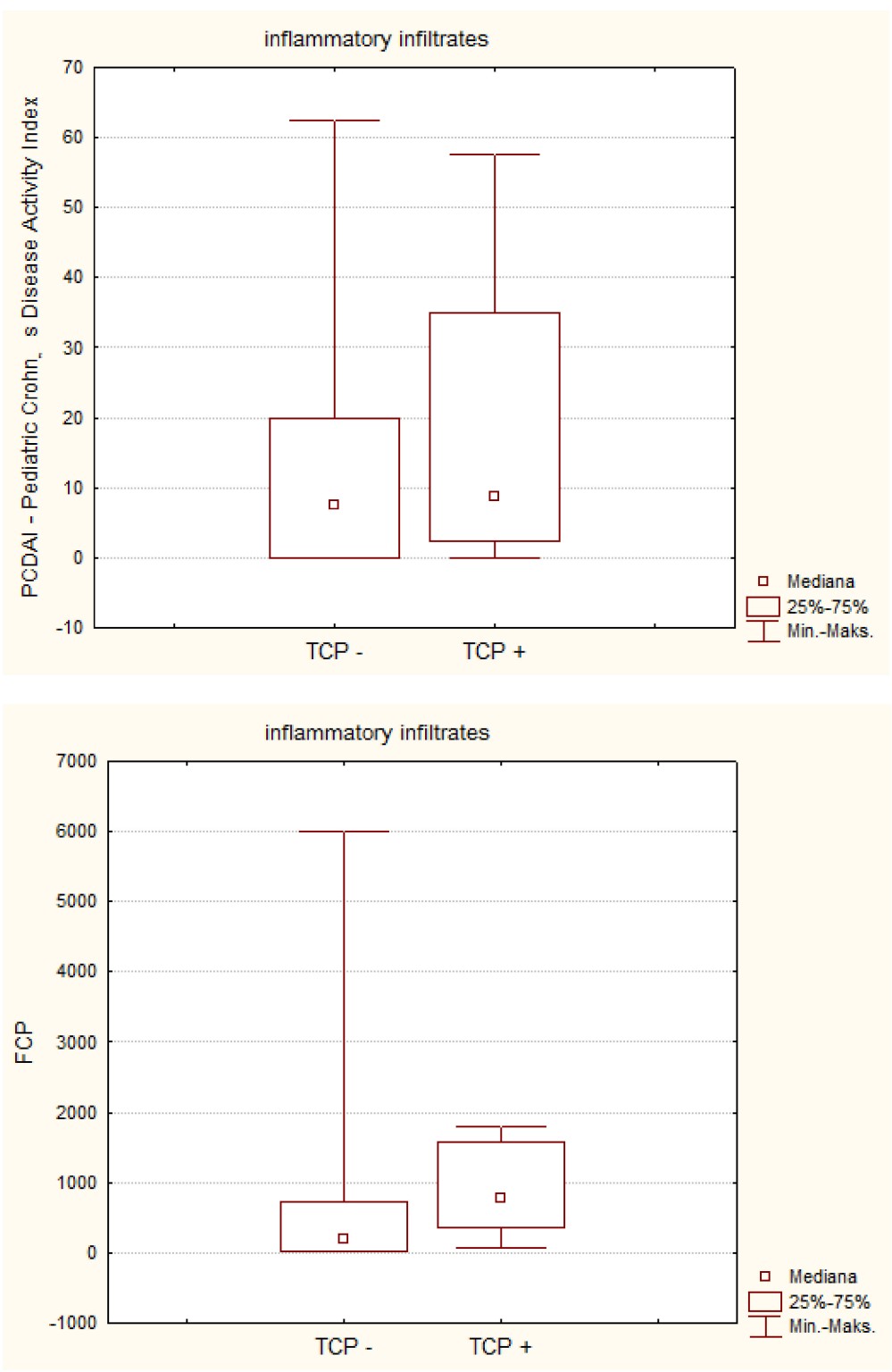

**Figure 2.** PCDAI and TCP in inflammatory infiltration in 57 patients included in the study; the higher the PCDAI score, the higher the TCP (*p* = 0.003).

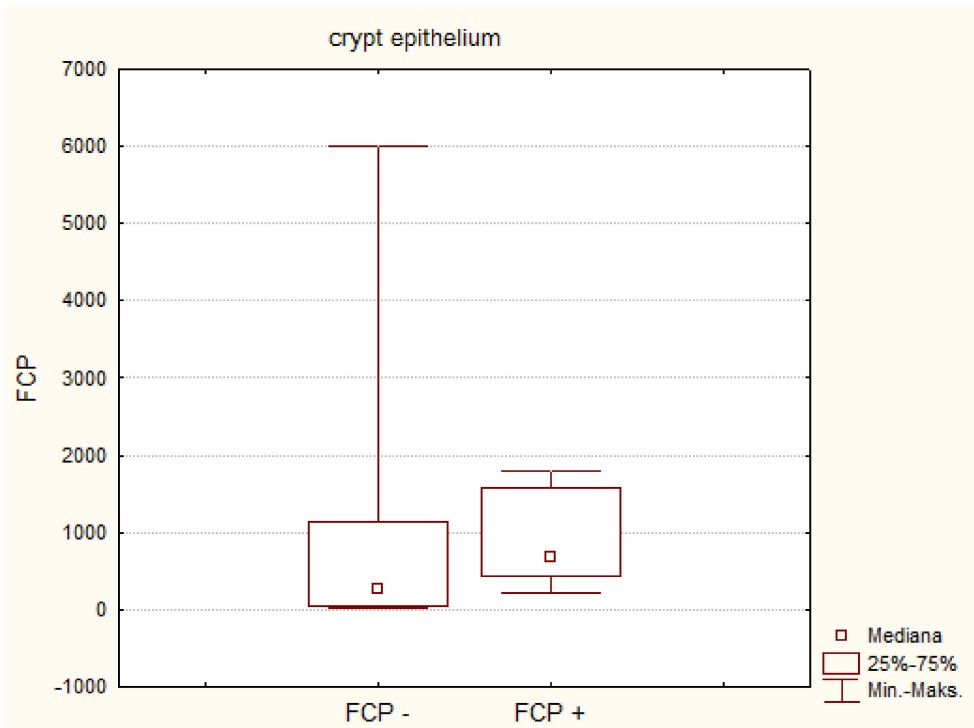

**Figure 3.** PCDAI and TCP in crypts in 57 patients included in the study; the higher the PCDAI, the higher the TCP in crypts (*p* = 0.025).

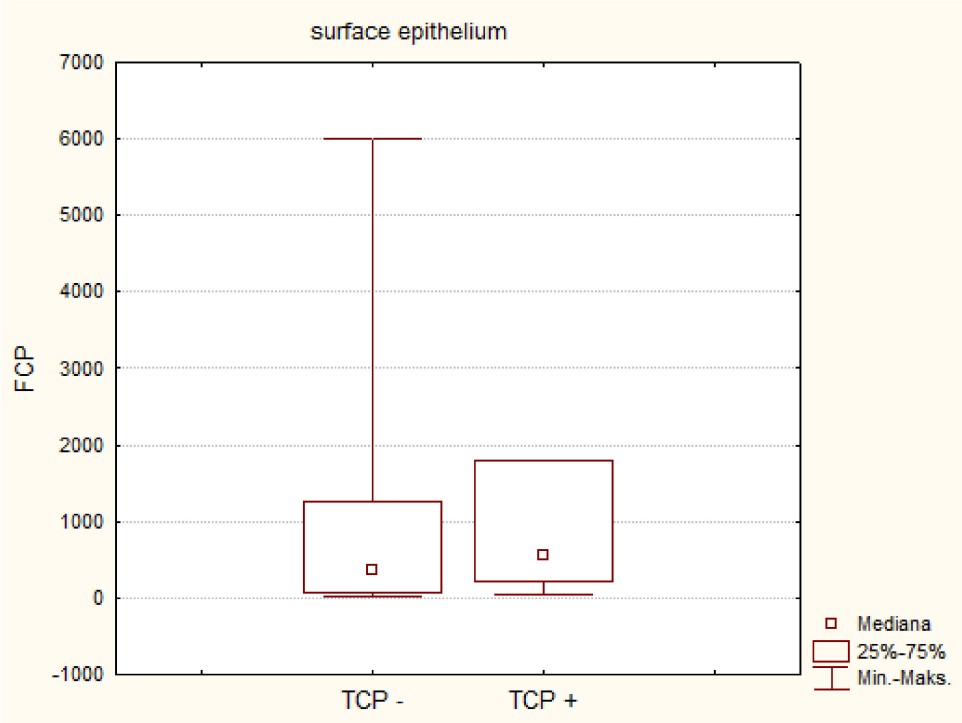

**Figure 4.** PCDAI and the surface TCP in 57 patients; the lower the PCDAI (milder disease), the higher the TCP in epithelium.

## 3. Discussion

Immunohistochemical detection of calprotectin in intestinal mucosa is not a commonly used method to evaluate the intensity of inflammation or predict disease recurrence. Thus, there are limited data on the usefulness of TCP in IBD.

Fabian et al. [17] evaluated the value of IHC detection of calprotectin in bowel mucosa in children with UC, and they found that TCP correlated well with microscopic scores, such as the Geboes and Nancy activity score. However, FCP and PUCAI appeared to be better predictors of unfavorable outcome in these patients [17]. This is consistent with both our results and many others confirming that FCP correlates well with clinical activity in IBD and has a good predictive value [19,20]. Although we did not observe a correlation between FCP and TCP, there was an association between TCP in inflammatory infiltration and in crypts, which seems logical.

Zarubova et al. [18] verified whether IHC detection of calprotectin in resection margins was useful in diagnosis of endoscopic recurrence in patients with CD, and they demonstrated that the predictive value of TCP was questionable, as it was negatively associated with endoscopic recurrence. They also observed that parameters such as CRP and FCP were better predictors at 6 months after resection and the presence of peritonitis [18]. In our study, we reported an association between TCP and histopathological features, such as inflammatory infiltration, but not with changes within epithelium. These outcomes, like our results, confirmed both the diagnostic and predictive value of FCP and the questionable value of TCP itself.

On the other hand, Fukunaga et al. [10] detected fecal, serum and tissue calprotectin in adults with IBD, and they reported that patients with both UC and CD had higher neutrophil and monocyte/macrophage calprotectin-positive cell expression levels compared with the control group. FCP was considered a reliable marker of disease activity [10]. These findings are partially concomitant with our outcomes, since we reported the correlation between clinical activity and TCP in inflammatory lesions and crypts but not in epithelium. Moreover, we even observed an inverse relationship between TCP in epithelium and disease activity.

A Chinese study that investigated the correlation between the disease activity of UC and the IHC distribution of calprotectin in colon mucosa and levels of FCP showed that both the TCP and the levels of FCP in UC were significantly correlated with the histological grades, respectively (r = 0.89, *p* = 0.0001; r = 0.849, *p* < 0.01), and the two parameters were well correlated (r = 0.90, *p* = 0.0007) [12]. Similarly, we observed the association between TCP and some histological features, as well as between FCP and disease activity.

Since calprotectin is an acute phase protein, and thus, its levels are increased in inflammatory diseases, we predicted that it would also be elevated in inflamed tissues. Our outcomes and the majority of sparsely available data demonstrated such correlation—between TCP and histopathological features or IBD activity. We assume that the pathophysiological explanation for our findings—the correlation between TCP in inflammatory lesions and crypts and PCDAI, but not between TCP in epithelium and PCDI—is that in CD, the whole intestinal wall is inflamed, not only the epithelium. The process begins and develops from the deepest intestinal parts. It seems logical that the level of TCP, which is the marker of intestinal destruction, is the highest in inflammatory lesions.

The limitation of our pilot study is the relatively small group sample and the semi-quantitative method for IHC. Additionally, differences in microscopic inflammatory intensity in endoscopic biopsies, which are not macroscopically visible, cannot be excluded. The number and size of tissue samples are important factors. The biopsy may come from the periphery of the most severe lesions, so the reaction does not reflect the actual activity of the disease in the tissue. Moreover, the tissue from endoscopic biopsies was examined retrospectively, and some clinical data are missing due to this retrospective design of the study.

However, the advantage of our study is a homogenous group of pediatric patients with CD who underwent standardized endoscopic and histopathological evaluation. Additionally, this is the first study assessing the usefulness of TCP in children with CD in Poland.

## 4. Patients and Methods

### 4.1. General Informations

This Is a Single-Center Retrospective Observational Cohort Study Evaluating Fecal and Tissue Calprotectin in Children with CD. The Values Were Correlated with Disease Activity and Histopathological Features.

### 4.2. Patients

4.2.1. Inclusion Criteria

1. Children aged 1 month–18 years.
2. Patients with confirmed diagnosis on histopathological examination at least 1 month prior to the study.
3. Patients receiving either nutritional treatment (Modulen IBD) or pharmacological treatment, including: oral corticosteroids, immunomodulators (azathioprine—AZA, methotrexate—MTX), biological agents (infliximab, adalimumab).

4.2.2. Exclusion Criteria

1. No confirmed diagnosis of CD or diagnosis of unclassified IBD or UC.
2. Disease demanding surgical intervention.
3. Patients who have not had endoscopy and histopathological examination.
More details are included in the Section 2 and in Table 1.

### 4.3. Fecal Samples

Children with CD were examined for fecal and tissue calprotectin. Their stool samples were frozen and stored at −80 °C. FCP was assessed by the competitive ELISA method (IDK® Calprotectin ELISA Kit, Immunodiagnostik AG, Bensheim, Germany), and the values were given in μg/g. According to the manufacturer, a median concentration <50 μg/g was estimated as the 'normal' value of FCP. Clinical data, such as PCDAI, SES-CD and important laboratory parameters, were also collected.

### 4.4. Histopathological Examination and CCP

In all biopsies, immunohistochemical (IHC) staining was performed using a monoclonal antibody against calprotectin (S100A8+S100A9 Calprotectin (27E10) GeneTex GTX17050) at a dilution of 1:100. The pattern of distribution of immunostaining was evaluated under a light microscope at X400 magnification. The semi-quantitative method was applied to assess calprotectin distribution in lamina propria, the crypts' epithelial cells and in inflammatory infiltrates, if they were present. The intensity of staining was assessed as follows:

- 0—negative staining;
- 1—faint and focal;
- 2—moderate;
- 3—strong.

### 4.5. Statistical Analysis

The intensity of staining was divided into 4 patterns. However, in our group, the majority of stainings were negative; therefore, for reliability of statistical analysis, we classified the IHC stainings either as positive (1) or negative (0). For a non-parametric measure of rank correlation, the U-test was used, while for the quantitative correlation, we used the Spearman test. Additionally, Pearson's correlation method was used to identify the negative, positive and neutral correlations between the analyzed parameters: calprotectin serum level vs. FCP patterns and Pediatric Crohn's Disease Activity Index vs. calprotectin patterns.

*4.6. Primary Outcome of the Study*

Assessment of TCP as a potential IHC marker of disease activity.

*4.7. Secondary Outcomes*

- Correlation between fecal and tissue calprotectin.
- Correlation between FCP and disease activity, and between TCP and disease activity.
- Correlation between FCP and TCP and histopathological features.

## 5. Conclusions

In our study, IHC stainings indicate that calprotectin is irregularly distributed in CD bioptates; it is present around the epithelium and crypts, as well as randomly within ECM. Therefore, it seems that IHC detection of calprotectin in bowel mucosa to assess disease behavior may be useful; however, this requires invasive methods, such as endoscopy.

FCP is still a gold-standard non-invasive biomarker in the diagnosis, monitoring and prognosis of IBD, and its levels correlated well with clinical activity in our study group.

**Author Contributions:** Conceptualization, E.S. and S.S.; methodology, E.S., J.K. and M.D.; software, M.D. and A.K.-W.; validation, M.D.; formal analysis, E.S., S.S. and J.K.; investigation, E.S. and S.S.; resources, E.S.; data curation, M.D and A.W.; writing—original draft preparation, E.S. and S.S.; writing—review and editing, E.S., S.S. and J.K.; visualization, E.S., S.S. and J.K.; supervision, J.K.; project administration, E.S.; funding acquisition, E.S. All authors have read and agreed to the published version of the manuscript.

**Funding:** This manuscript was financially supported by the Institutional grant M43/2019.

**Institutional Review Board Statement:** The study was conducted in accordance with the Declaration of Helsinki, and the protocol was approved by the local bioethical committee—resolution 49/KBE/2019 (6 November 2019).

**Informed Consent Statement:** This paper doesn't contain case studies or other examples using individual people with identifying information.

**Data Availability Statement:** All data are contained within the article.

**Conflicts of Interest:** The authors declare no conflict of interest.

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
