# Peer review of "The Usefulness of Tissue Calprotectin in Pediatric Crohn’s Disease—A Pilot Study"

_gastrointestdisord, doi:10.3390/gidisord5010003_

Round 1

Reviewer 1 Report (Previous Reviewer 2)

The authors improved the manuscript following my comments. Nevertheless, I still think that legends titles for Figures 2-4 should be corrected. The figures do not present correlation but are bar charts presenting changes in examined parameters between two groups. The authors should use parametric or non-parametric statistical tests to determine statistical significance on those figures. Also the figure legends should contain information on the number of patients in each grop.

Author Response

Thank You for your valuable comments

Please find the response below

Rev 1

The authors improved the manuscript following my comments. Nevertheless, I still think that legends titles for Figures 2-4 should be corrected. The figures do not present correlation but are bar charts presenting changes in examined parameters between two groups. The authors should use parametric or non-parametric statistical tests to determine statistical significance on those figures. Also the figure legends should contain information on the number of patients in each grop

We have changed the descriptions of the figures – marked in red

Figure 1. The level of FCP and clinical disease activity (PCDAI) of 57 patients included in the study: the higher the PCDAI score (more active disease), the higher the FCP level was (r-Pearson: 0.31, r-spearman – 0.186, p- 0.16)

Figure 2. PCDAI and TCP in inflammatory infiltration in 57 patients included in the study: the higher the PCDAI score, the higher the TCP (p=0.003)

Figure 3. PCDAI and TCP in crypts in 57 patients included un the study: the higher the PCDAI, the higher the TCP in crypts (p=0.025).

Fig 4.PCDAI and the surface TCP in 57 patients - the lower the PCDAI (milder disease), the higher the TCP in epithelium was. 

Reviewer 2 Report (Previous Reviewer 3)

The article reads smoother.
Only major problem is the lumping of CD with IBD The study is focused on CD
Abstract -Remove line 27-30  - Allow the data to speak for itself
Introduction - The reference is date and as such is not appropriate. The focus needs to be on education.
Lumping CD with IBD is a disservice. We know the pathogenesis of CD and do not have the slightest idea as to the causation of UC. What they share in common is the induction of cytotoxic destruction of the gastrointestinal mucosa with resultant inflammation which is measurable using TCP. Simply state that CD is an immune-mediated disease and cite modern references. Yes, genetics plays a role , but it does so in virtually every infectious disease entity. Throwing CD into the garbage bag of IBD does little to advance science. Add correct referenceS
Line 66 . Consider deleting Therefore. .. and providing added insight It weakens the statement

Author Response

Thank You for your valuable comments

Please find the response below

Rev 2

Only major problem is the lumping of CD with IBD The study is focused on CD
Abstract -Remove line 27-30  - Allow the data to speak for itselfBecause line 27-30
Introduction - The reference is date and as such is not appropriate. The focus needs to be on education.

Could You please specify which line exactly You recommend to delete? Becsue line 27-30 is conclusion…

Lumping CD with IBD is a disservice. We know the pathogenesis of CD and do not have the slightest idea as to the causation of UC. What they share in common is the induction of cytotoxic destruction of the gastrointestinal mucosa with resultant inflammation which is measurable using TCP. Simply state that CD is an immune-mediated disease and cite modern references. Yes, genetics plays a role , but it does so in virtually every infectious disease entity. Throwing CD into the garbage bag of IBD does little to advance science. Add correct referenceS

We changed the introduction sentence into:

Crohn's disease (CD) is a chronic gastro-intestinal condition with impaired immunological background.[i] Diagnosis of CD is based on clinical symptoms, laboratory parameters, and histopathological examination of the endoscopic intestinal biopsies.[ii],[iii]
Line 66 . Consider deleting Therefore. .. and providing added insight It weakens the statement

We deleted therefore – marked in red:      The aim of this study was to evaluate TCP as a biomarker providing added insight for pediatric patients with CD.

[i] Michielan A.; D'Incà R. Intestinal Permeability in Inflammatory Bowel Disease: Pathogenesis, Clinical Evaluation, and Therapy of Leaky Gut. Mediators Inflamm. 2015;2015:628157.

[ii] Gomollón F.; Dignass A.; Annese V et al; ECCO 3rd European Evidence-based Consensus on the Diagnosis and Management of Crohn's Disease 2016: Part 1: Diagnosis and Medical Management. J Crohns Colitis. 2017;11(1):3-25.

[iii] Magro F.; Gionchetti P.; Eliakim R et al; European Crohn’s and Colitis Organisation [ECCO]. Third European Evidence-based Consensus on Diagnosis and Management of Ulcerative Colitis. Part 1: Definitions, Diagnosis, Extra-intestinal Manifestations, Pregnancy, Cancer Surveillance, Surgery, and Ileo-anal Pouch Disorders. J Crohns Colitis. 2017;11(6):649-670.

This manuscript is a resubmission of an earlier submission. The following is a list of the peer review reports and author responses from that submission.

Round 1

Reviewer 1 Report

In the present manuscript, the authors attempt to investigate the role of calprotection in the tissues of pediatric patients with Crohn's disease. For this purpose, they did a patient study of 57 children who had stool samples and biopsies taken. Then the calprotectin level was determined with the help of an ELISA and with the help of IHC staining the distribution patterns in the biopsy tissue.

First of all, it is striking that the authors do not define any criteria for the inclusion or exclusion of patients in their cohort. Items such as treatment, clinical data, endoscopic scores and biopsy sites should be included.

In Material & Methods, the parameter given is the intensity of staining as negative, weak, moderate and strong. What does this property refer to? To the number of positive cells or area of brown colour?

In Figures 1 - 8 incomprehensible labels are given as axis labels or headings. Please change these.

A representation of the box-wisker plots is unfortunate, as e.g. in Figure 1 huge error bars can be seen and the negative bar has been cut off. Here, a representation as a dot-box plot would be better.

Figure 8 shows a correlation of inflammatory activity with the amount of calprotectin, which has already been shown in numerous publications. Additional data would be necessary here, such as a correlation of fecal cal against tisseu cal. Another parameter would be to analyse the number of calprotectin-producing cells such as neutrophils and monocytes/macrophages against the level of tissue calprotectin.

Figure A and B shows an IHC stain of a biopsy, once at 200x and once at 100x magnification. This is unusual as normally a figure would first show an image of healthy colon tissue compared to inflamed colon tissue. The negative control is also missing in this figure. The orientation of the figure is unclear. Where is the surface? Where is the muscle layer and where are the crypts? Why was the image enlarged once at 100x or 200x and not directly at 400x so that the cell structures can be seen better?

The references are oddly numbered, once in roman numerals and sometimes in arabic numerals. Please number them consistently.

Thematically, the study is not very new. Calprotectin enters the intestinal lumen as a component of polymorphonuclear granulocytes as part of an inflammatory process of mucosal epithelial cells and can thus be seen as a marker of a cellular inflammatory process and easily determined in stool. Among the faecal markers, calprotectin is considered to have the best diagnostic potential. Already in 2015, Sipponen et Kolho postulated that calprotectin is an objective biomarker of mucosal healing that correlates with disease activity in MC and CU and can help differentiate between organic and non-organic intestinal diseases (Sipponen and Kolho 2015 Aliment Pharmacol Ther, 28:1221). The focus of the manuscript presented here is now to compare histology findings with calprotectin levels from stool and biopsies to evaluate whether calprotectin can be used as a surrogate parameter of inflammatory activity in inflammatory bowel disease.

inflammatory bowel diseases. Here, unfortunately, further markers such as a correlation of calprotectin with the serological inflammatory markers CRP and number of leucocytes or endoscopy values are missing, which could allow a conclusive evaluation.

Overall, the manuscript has considerable qualitative deficiencies in terms of presentation. All figures need to be completely revised. In terms of content, more markers are missing that could support the thesis of the study. For this reason, the manuscript can only be rejected so that the authors can revise it comprehensively.

Reviewer 2 Report

The study by SzymaÅ„ska et al. Entitled “The usefulness of tissue calprotectin in pediatric Crohn’s dis- 2 ease – a pilot study” compare tissue calprotectin level in stool samples of pediatric patients. The authors analyse 57 patients and tried to correlate the TCP level with disease activity. The approach is interesting and the study group is big enough to draw conclusions; however, the study must be extensively corrected.

Major comments:

1.     Data presentation- all figures and figures legends must be translated to English. Units should be added to OY axis.

2.     The authors measured TCP level in two group of patients (with different disease activity), therefore only one column bar should be presented. The correlation is depicted in figure 8. In fact, the authors present three data: 1. FCP level in CD patients with different disease activity. 2. Correlation between FCP and disease activity, 3. IHC stainings of calprotectin.

3.     If the authors show IHC stainings than representative stainings from patients with different disease activity should be added. Moreover, a negative control should be included.

4.     The authors may use approach described by Kopaz et al. (doi: 10.3390/ijms23116175) to quantify the IHC signal, than it could be e.g., correlated with the level of FCP. It would increase the value of presented data as the number of analysed patients is relatively high.

5.     How the authors determined that “TCP in inflammatory infiltration and in crypts correlated with clinical activity of CD”. It is not presented in the results. To assess that TCP correlates with inflammatory infiltration a constraining with lymphocytes markers should be done. Provided IHC stainings indicate that calprotectin is irregularly distributed in CD bioptates, it is present around the epithelium and crypts, as well as randomly within ECM. This random localization may be associated with lymphocytes but it has to be convincingly presented.

6.     The references presentation should be corrected and adjusted to the journal requirements.

Reviewer 3 Report

This is a measurement paper and as such should be so constructed and properly focused

Abstract:

recommend deleting line14-14

recommend restating line 15-18 the correlation of FCP to clinical activity is well established. The question being posed does evaluation of TFC provide added insight for pediatric patents with Crohn’s disease

the problem with biopsy data is sample error -demonstration that it may correlated with a histological site of inflammation is of some value

line 25-26 should be the abstracts conclusion

Introduction:

Lead sentence is off the mark. CD has been documented to be an immune-mediated disease (Med. Hypothesis 85:878-881. Dx.doi.org/10.1016/j.mehy.2015.09.019; Gastrointest. Discord. 3:138-141.doi.org/10.3390/gdiscord30300150. When mononuclear cells from an individual with naïve CD are incubated then with MAP antigenic array appropriate for the region TNNF-alpha and IL1 are produced.

What UC is up for debate. Both share the ability to cause inflammation of the gastrointestinal mucosa. They do not share a common etiology. This paper needs to focus of FCP quantitation in CD pediatric patents. Recommend deletion of 34-39 and 49-53

Line 55 Delete However in contrary to FP

Rephrase line 62-63

Rewrite Discussion

Begin discussion with 195-201

Focus on pediatric CD in adults is a much more complicated disease that could impact of FCP (Adv. Res. Gastroenterol Hepatol. 5:1-2. Doi.10.19080?ARGH 2017.05555675) Delete or change 172-178. Restrict discussion of IBD or UC to explanation the role of FCP in disease. 

Being a retrospective study, small number of observations, sample bias introduced by biopsy site selection, and the need for editorial assistance in conversion of text to English would normally be sufficient to negate publication.

To do so, would miss the bigger issue of encouraging young investigators in under published countries to question what they are doing. I have not read many papers from Poland. There is some solid observational data which expanded with more educational discussion about FCP would warrant publication.

I recommend that these authors be invited to rewrite with the journal’s assistance an educational paper on FCP in pediatric CD patients.

This is a measurement paper and as such should be so constructed and properly focused

Abstract:

recommend deleting line14-14

recommend restating line 15-18 the correlation of FCP to clinical activity is well established. The question being posed does evaluation of TFC provide added insight for pediatric patents with Crohn’s disease

the problem with biopsy data is sample error -demonstration that it may correlated with a histological site of inflammation is of some value

line 25-26 should be the abstracts conclusion

Introduction:

Lead sentence is off the mark. CD has been documented to be an immune-mediated disease (Med. Hypothesis 85:878-881. Dx.doi.org/10.1016/j.mehy.2015.09.019; Gastrointest. Discord. 3:138-141.doi.org/10.3390/gdiscord30300150. When mononuclear cells from an individual with naïve CD are incubated then with MAP antigenic array appropriate for the region TNNF-alpha and IL1 are produced.

What UC is up for debate. Both share the ability to cause inflammation of the gastrointestinal mucosa. They do not share a common etiology. This paper needs to focus of FCP quantitation in CD pediatric patents. Recommend deletion of 34-39 and 49-53

Line 55 Delete However in contrary to FP

Rephrase line 62-63

Rewrite Discussion

Begin discussion with 195-201

Focus on pediatric CD in adults is a much more complicated disease that could impact of FCP (Adv. Res. Gastroenterol Hepatol. 5:1-2. Doi.10.19080?ARGH 2017.05555675) Delete or change 172-178. Restrict discussion of IBD or UC to explanation the role of FCP in disease. 

Being a retrospective study, small number of observations, sample bias introduced by biopsy site selection, and the need for editorial assistance in conversion of text to English would normally be sufficient to negate publication.

To do so, would miss the bigger issue of encouraging young investigators in under published countries to question what they are doing. I have not read many papers from Poland. There is some solid observational data which expanded with more educational discussion about FCP would warrant publication.

I recommend that these authors be invited to rewrite with the journal’s assistance an educational paper on FCP in pediatric CD patients.

This is a measurement paper and as such should be so constructed and properly focused

Abstract:

recommend deleting line14-14

recommend restating line 15-18 the correlation of FCP to clinical activity is well established. The question being posed does evaluation of TFC provide added insight for pediatric patents with Crohn’s disease

the problem with biopsy data is sample error -demonstration that it may correlated with a histological site of inflammation is of some value

line 25-26 should be the abstracts conclusion

Introduction:

Lead sentence is off the mark. CD has been documented to be an immune-mediated disease (Med. Hypothesis 85:878-881. Dx.doi.org/10.1016/j.mehy.2015.09.019; Gastrointest. Discord. 3:138-141.doi.org/10.3390/gdiscord30300150. When mononuclear cells from an individual with naïve CD are incubated then with MAP antigenic array appropriate for the region TNNF-alpha and IL1 are produced.

What UC is up for debate. Both share the ability to cause inflammation of the gastrointestinal mucosa. They do not share a common etiology. This paper needs to focus of FCP quantitation in CD pediatric patents. Recommend deletion of 34-39 and 49-53

Line 55 Delete However in contrary to FP

Rephrase line 62-63

Rewrite Discussion

Begin discussion with 195-201

Focus on pediatric CD in adults is a much more complicated disease that could impact of FCP (Adv. Res. Gastroenterol Hepatol. 5:1-2. Doi.10.19080?ARGH 2017.05555675) Delete or change 172-178. Restrict discussion of IBD or UC to explanation the role of FCP in disease. 

Being a retrospective study, small number of observations, sample bias introduced by biopsy site selection, and the need for editorial assistance in conversion of text to English would normally be sufficient to negate publication.

To do so, would miss the bigger issue of encouraging young investigators in under published countries to question what they are doing. I have not read many papers from Poland. There is some solid observational data which expanded with more educational discussion about FCP would warrant publication.

I recommend that these authors be invited to rewrite with the journal’s assistance an educational paper on FCP in pediatric CD patients.